# Disparities in Survival Outcomes among Racial/Ethnic Minorities with Head and Neck Squamous Cell Cancer in the United States

**DOI:** 10.3390/cancers15061781

**Published:** 2023-03-15

**Authors:** Sujith Baliga, Vedat O. Yildiz, Jose Bazan, Joshua D. Palmer, Sachin R. Jhawar, David J. Konieczkowski, John Grecula, Dukagjin M. Blakaj, Darrion Mitchell, Christina Henson, Kenneth Hu, Kosj Yamoah, Mauricio E. Gamez

**Affiliations:** 1Department of Radiation Oncology, Ohio State University Wexner Medical Center, Columbus, OH 43210, USA; 2Department of Radiation Oncology, Oklahoma University Health Stephenson Cancer Center, Oklahoma City, OK 73104, USA; 3Department of Radiation Oncology, New York University School of Medicine, New York, NY 10016, USA; 4Department of Radiation Oncology, Moffitt Cancer Center, Tampa, FL 33612, USA; 5Department of Radiation Oncology, Mayo Clinic, Rochester, MN 55902, USA

**Keywords:** racial disparities, head and neck cancer, squamous cell

## Abstract

**Simple Summary:**

Racial disparities in head and neck cancers contribute to mortality in racial/ethnic (R/E) minorities, but the specific risk factors associated with inferior survival are poorly understood. The aim of our retrospective study was to describe and report clinical and treatment-related survival outcomes by specific racial and ethnic groups, and determine if specific demographic, clinical, and socioeconomic factors could be associated with inferior outcomes among R/E minorities. In this study, we demonstrated that Black race was independently associated with worse overall survival rates across multiple head and neck disease subsites. We also showed that Black patients presented with more advanced stage of disease and had longer total treatment package times compared to White patients. We conclude that more research is needed to understand the biological basis for the worse outcomes identified in our results, after controlling for the social determinants of health.

**Abstract:**

Background: Racial/ethnic (R/E) minorities with head and neck squamous cell carcinoma (HNSCC) have worse survival outcomes compared to White patients. While disparities in patient outcomes for R/E minorities have been well documented, the specific drivers of the inferior outcomes remain poorly understood. Patients and Methods: This was a population-based retrospective cohort study that analyzed HNSCC patients using the National Cancer Database (NCDB) from 2000–2016. Patient outcomes were stratified by R/E groups including White, Black, Hispanic, Native American/Other, and Asian. The main outcome in this study was overall survival (OS). Univariate time-to-event survival analyses were performed using the Kaplan–Meier product limit estimates and the log-rank test to evaluate the differences between strata. Results: There were 304,138 patients with HNSCC identified in this study, of which 262,762 (86.3%) were White, 32,528 (10.6%) were Black, 6191 were Asian (2.0%), and 2657 were Native American/Other (0.9%). Black R/E minorities were more likely to be uninsured (9% vs. 5%, *p* < 0.0001), have Medicaid insurance (22% vs. 8%, *p* < 0.0001), be in a lower income quartile (<30,000, 42% vs. 13%, *p* < 0.0001), have metastatic disease (5% vs. 2%, *p* < 0.001), and have a total treatment time 6 days longer than White patients (median 107 vs. 101 days, *p* < 0.001). The 5-year OS for White, Black, Native American/Other, and Asian patients was 50.8%, 38.6%, 51.1%, and 55.8%, respectively. Among the oropharynx HNSCC patients, the 5-year OS rates in p16+ White, Black, and Asian patients were 65.7%, 39.4%%, and 55%, respectively. After a multivariate analysis, Black race was still associated with an inferior OS (HR:1.09, 95% CI: 1.03–1.15, *p* = 0.002). Conclusions: This large cohort study of HNSCC patients demonstrates that Black race is independently associated with worse OS, in part due to socioeconomic, clinical, and treatment-related factors.

## 1. Introduction

According to the U.S. Census Bureau, racial/ethnic minorities comprise 37% of the United States population and are expected to make up 57% of the population by 2060. Cancer outcomes, and particularly death rates among minority populations, are typically poorer than in the White population [1]. Head and neck cancers (HNCs) are of particular interest given the rising incidence and complexity of their treatment. According to recent cancer statistics, oral cavity and pharynx cancers will account for 54,540 new cases in 2023 and 11,580 estimated deaths in the United States [2]. On a global scale, HNCs accounted for over 700,000 new cases and 444,347 deaths in 2020 [3]. In addition, cancer-independent life expectancy is 6.5 years shorter than that expected for HNC patients [4]. Finally, the suicide rate among HNC survivors is nearly twice that of other cancers [5]. Research and technological advances in the delivery of surgery, chemotherapy, and radiotherapy, as well as the decline in smoking and the discovery of the prognostic impact of the Human Papilloma Virus (HPV) on the survival of oropharynx cancer patients, has continued to improve disease outcomes over the last decade [6]. However, studies demonstrating these findings have typically not focused on or adequately represented minority patients, and gaps in cancer survival continue to exist for racial/ethnic (R/E) minorities. For example, outcome estimates reported by the American Cancer Society reflect up to an 18% survival difference between Black and White patients with oral cavity and pharynx tumors [7]. Other analyses have also shown inferior survival rates for Black patients even after accounting for other demographic and clinical factors [8,9,10,11]. Despite the increased awareness of disparities in cancer outcomes in head and neck squamous cell carcinoma (HNSCC), the specific drivers of inferior outcomes among these groups have been underreported and are poorly understood. In addition, survival outcomes of other racial/ethnic groups, such as Asian, Native American, and Hispanic are not as well documented in the literature. Defining the specific drivers of R/E disparities may facilitate the development of specific socioeconomic strategies and clinical interventions with an aim to narrow these gaps. Therefore, we utilized a large national U.S. database to comprehensively describe and report clinical and treatment-related survival outcomes by specific race and ethnic group, amongst different HNC subsites, along with the associated demographic and socioeconomic factors that could be contributing to these outcomes.

## 2. Materials and Methods

The National Cancer Database (NCDB) is a joint project of the Commission on Cancer (CoC) of the American College of Surgeons and the American Cancer Society. The CoC’s NCDB and the participating hospitals are the source of the de-identified data used herein; they have not verified and are not responsible for the statistical validity of the data analysis or the conclusions derived by the authors. Data within the NCDB include basic demographics, cancer staging, comorbidities, therapies delivered during the first course of treatment, and OS. The NCDB does not capture disease recurrence or salvage therapies.

The NCDB was queried to identify patients with newly diagnosed American Joint Committee on Cancer (AJCC) 7th edition-clinical T1-T4, N0-N3, and M0-M1 HNSCC of the oropharynx, oral cavity, nasal cavity, pharynx (NOS), hypopharynx, nasopharynx, and larynx between 2000–2016. Primary site SEER codes 8070–8078 were applied and used to include only patients with a diagnosis of squamous cell carcinoma. Patients treated with any treatment modality (surgery, radiation therapy, or chemotherapy), or a combination of them, were included in the final analysis. Patients were excluded if they had an unknown clinical T or N stage, if radiation was given to an area other than the head and neck, or if details regarding the administration of chemotherapy or surgery were not clear or were unavailable. To ascertain HPV status, SEER “CS-Site Specific Factor 10” was used to identify high-risk HPV subtypes. Patients with codes 10 (HPV positive for low-risk types only), 20 (HPV positive for specified high-risk type(s) other than 16 or 18), 30 (HPV positive for high-risk type 16), 40 (HPV positive for high-risk type 18), 50 (HPV positive for high-risk types 16 and 18), 60 (HPV positive for high-risk types, not otherwise specified, high risk type(s) not stated), and 70 (HPV positive, NOS, risk and type(s) not stated) were considered to be HPV positive. Figure 1 shows the selection criteria used to determine our final cohort. After our final cohort was identified, patients were stratified according to four main R/E groups: White, Black, Native American/Other, and Asian. Further stratification was also carried out based on ethnicity including the Black-Hispanic and White-Hispanic groups.

### Statistical Analysis

Descriptive statistics were used to summarize patient characteristics, using mean ± standard deviation or median (interquartile range, IQR) for continuous variables and frequency counts with percentages for categorical variables. Analyses for proportions of categorical variables were performed using a Chi-Square or Fisher’s Exact test. The analysis of differences between race in continuous variables were tested using analysis of variance (ANOVA) or Kruskal–Wallis tests as appropriate according to distribution of residuals. Univariate time-to-event survival analyses were performed using Kaplan–Meier product limit estimates and the log-rank test to evaluate the differences between strata. A Cox proportional hazards regression model was used to assess the association between covariates and the risk of death or overall survival (OS). Variables included in the multivariable analysis were race, age, sex, ethnicity, insurance status, income, clinical stage, receipt of radiation, receipt of surgery, Charlson–Deyo Co-Morbidity Index score, presence of metastasis, HPV status, total radiation dose, and distance from treatment center. Chemotherapy and radiation (RT) variables were not included together in any multivariable model to avoid multi-collinearity issues due to a high correlation between these two variables. All statistical tests were 2-sided and evaluated at the α = 0.05 type-I error rate and deemed significant at *p* < 0.05. SAS software (version 9.4; SAS Institute Inc., Cary, NC, USA) was used in all analyses.

To balance the covariates between the subjects in the racial groups, a propensity score greedy nearest-neighbor matching was performed using the SAS PSmatch procedure. We performed three pairwise matchings in the R/E groups: Asian vs. White, Asian vs. Black, and Black vs. White. Propensity scores were developed through logistic regression, with age, sex, insurance, income, clinical stage group, Charlson–Deyo Comorbidity Index, surgery, HPV status, and chemotherapy as independent variables and race as the dependent variable. A Cox proportional hazards analysis was then performed to determine whether overall survival in patients was different between the R/E groups. All statistical analyses were performed using SAS version 9.4 (SAS Institute, Cary, NC, USA).

## 3. Results

### 3.1. Demographic and Clinical Comparisons

After excluding patients who did not meet the selection criteria (Figure 1), we identified 304,138 patients with a diagnosis of HNSCC, of which 262,762 (86.3%) were White, 32,528 (10.6%) were Black, 6191 were Asian (2.0%), and 2657 were Native American/Other (0.9%). A total of 11,905 patients were categorized as being from Hispanic ethnicity, of which 11,036 (92.3%) were White-Hispanic. Table 1 shows baseline differences between clinical and demographic characteristics by race and ethnicity. Compared to White patients, Black patients were more likely to be younger at the time of diagnosis with a mean of 64 vs. 61.2 years (*p* < 0.0001), to be uninsured (9% vs. 5%, *p* < 0.0001) or to have Medicaid insurance (22% vs. 8%, *p* < 0.0001), to be in the lower income quartile (<30,000$, 42% vs. 13%, *p* < 0.0001), to have Stage IV disease (39% vs. 29%, *p* < 0.0001), to present with distant metastatic disease (5% vs. 2%, *p* < 0.0001), to have a longer surgery to radiation start (SRT) interval of >6 weeks (64% vs. 58%), and to have a total treatment time delay, defined as time from diagnosis to completion of all treatment, 6 days longer than White patients (median 107 vs. 101 days, *p* < 0.001). Furthermore, Black patients were less likely to be diagnosed with HPV-positive oropharyngeal carcinoma (11% vs. 6%, *p* < 0.001) and less likely to be treated with primary surgery (62% vs. 44%, *p* < 0.0001).

### 3.2. Survival Analysis

The 5-year OS for White, Black, Native American/Other, and Asian R/E groups, among all head and neck subsites, was 50.8%, 38.6%, 51.1%, and 55.8%, respectively (Figure 2A). When comparing White vs. non-White patients for all subsites, non-White patients had an inferior 5-year OS of 42.1% vs. 51.2% (Figure 2B; *p* < 0.0001). The 5-year OS for Black-Hispanic vs. Black non-Hispanic patients was 49% vs. 38% (Figure 2C, Log-rank *p* < 0.0001). The 5-year OS for White Hispanic vs. White non-Hispanic was 51.5% versus 51.4% (*p* = 0.511). Across all clinical AJCC stage groups, Asian patients had the best 5-year OS and Black patients had the worst OS (Figure 2D–G). Black patients had an inferior 5-year OS across multiple head and neck subsites, including the hypopharynx, larynx, nasal cavity, nasopharynx, pharynx, and oral cavity (Figure 3, All Log rank *p* < 0.001). Among the oropharynx HNSCC patients, the 5-year OS rates in p16+ White, Black, and Asian patients were 65.7%, 39.4%%, and 55%, respectively (Appendix A). For HPV-negative patients, the 5-year OS among White and Black patients was 36.5% vs. 13.8% (Appendix A, *p* < 0.0001). Among Black HNSCC patients, survival was improved for patients with higher incomes, those who had insurance, and those who received surgery (Appendix A).

The multivariate analysis showed that after adjusting for multiple clinical and demographic predictors including age, sex, race, ethnicity, income, stage, surgical treatment, co-morbidity score, metastatic disease, HPV status, and travel distance, Black race was still associated with an inferior OS (HR:1.09, 95% CI: 1.03–1.15, *p* = 0.002) compared to White race (Table 2).

Propensity score matching demonstrated similar findings to the above, as Black patients had a higher risk of death (HR: 1.13, 95% CI: 1.09–1.16) compared to White and Asian patients (Appendix A).

### 3.3. Analysis of Treatment Time

In this cohort, the time interval from surgery to the start of RT (SRT) was a mean of 57.6 Days (IQR 38–68 days) in non-White patients versus 53.6 days (IQR 35–62 days) in White patients (*p* < 0.0001). Among White, Black, Native American/Other, and Asian patients, the median SRT was 47, 50, 51, and 49 days, respectively. Among all surgically treated patients, the 5-year OS for an SRT interval >6 weeks versus ≤6 weeks was 49% vs. 57%, respectively (log-rank, *p* < 0.0001). Among the R/E groups, both White (log-rank, *p* < 0.001) and Black patients (*p* = 0.04) had an inferior OS for an SRT > 6 weeks.

## 4. Discussion

To our knowledge, this is the largest study to evaluate the impact of race and ethnicity on survival in patients with HNSCC by specific anatomic subsite. Using a large national database, we demonstrate inferior survival outcomes among non-white HNSCC, specifically Black Hispanic and Black non-Hispanic, which seem to be primarily influenced by socioeconomic factors such as race, insurance status, and income level. The study has several novel findings including the finding that differences in survival by race persist for the head and neck subsites, the improved survival for Asian patients compared to Black and White patients, and the poor survival of HPV-positive Black OPSCC patients who have a similar survival to HPV-negative OPSCC White patients. Our work builds upon the prior literature in racial and ethnic disparities in HNSCC patients by demonstrating consistent differences in patients for nearly all subsites of HNSCC. Furthermore, our study identifies specific clinical and demographic predictors of survival, such as clinical stage, treatment modality, income, and insurance status which may help national health policy makers to develop strategies to reduce disparities by acknowledging and addressing these gaps.

As in previous reports [12,13,14,15], we identified significant disparities in socioeconomic and health insurance coverage among Black HNSCC patients. Similarly, a SEER database analysis from Taylor et al., showed poorer outcomes in Black patients and demonstrated that socioeconomic status and health insurance were associated with head and neck cancer-specific survival and stage at presentation [16]. These findings are not unique to HNSCC patients. For example, Halpern et al., demonstrated that uninsured patients, Medicare-insured patients and ethnic minorities were more likely to present with an advanced stage of disease among the twelve cancer sites evaluated [17]. Our analysis builds upon the aforementioned studies by demonstrating consistent decrements in survival for Black patients across multiple anatomic subsites in the head and neck, and also demonstrates important survival decrements due to the prolongation of treatment time.

It is well known that HPV-mediated oropharynx SCC (HPV-OPSCC) has been associated with improved outcomes compared to HPV-negative OPSCC, predominantly in White patients [6,18,19,20]. An interesting and concerning finding in this study is that the survival of Black HPV-OPSCC patients was nearly equivalent to the HPV-negative White cohort. This is certainly intriguing as this is a specific patient population where de-intensification strategies are rigorously being evaluated in prospective trials due to their associated excellent oncologic outcomes [19,21,22,23]. Further research, therefore, is important for ensuring that such trials are appropriate for patients of all demographic groups. A smaller retrospective study from the Carolina Head and Neck Cancer Epidemiology Study cohort demonstrated inferior survival outcomes in Black HPV-OPSCC patients, even after adjusting for demographic, clinical, and socioeconomic variables [24]. In another study using the SEER database, being non-White was associated with a worse cancer-specific mortality in HPV-OPSCC patients compared to being White [25]. Potential reasons for worse outcomes in this population include the difficulty in accessing medical care for non-White patients, a lower household income, the later detection of disease, and unknown genomic/biologic differences.

The clinical significance of the prolonged time to the initiation of postoperative radiation therapy in surgically resected head and neck patients, with an associated negative impact on survival, has been previously described [26,27,28]. Naghavi and colleagues evaluated outcomes in 1802 non-metastatic head and neck cancer patients and demonstrated that Black patients were more likely to have had a longer time between diagnosis and treatment initiation [29]. Other studies in breast cancer patients show that delays in treatment initiation and prolonged treatment duration were more likely to occur in Black patients versus White patients. In addition, both a low socioeconomic status and more barriers to care, such as being uninsured, job loss, financial, and transportation issues, were also associated with a prolonged treatment duration [30]. In our study, the surgery to the RT interval was 3 days longer in Black patients and 4 days longer in Native American/Other patients, and had a negative impact on survival. The reason for this delay is not particularly clear as the NCDB does not provide this information; however, some potential reasons could include a delay in referral to the radiation oncology provider, a delay in dental clearance, and underlying transportation/socioeconomic issues. An equally concerning finding in our study is that Black patients were less likely to receive primary surgical resection in the upfront management of their HNSCC, compared to White patients. This could be in partly due to proportional differences in cancer subsites, with the Black population having a higher incidence of larynx cancer, which is often managed with definitive radiation, and a lower incidence of oral cavity cancer, which is typically managed surgically. However, given the critical importance of surgery, especially in oral cavity cancers, future studies should evaluate whether the differences in resection rates are due to clinical risk factors, such as the advanced stage and inoperability of the cancer, or due to other intangibles, such as socioeconomic issues or provider bias.

Another interesting finding in this study was that Black HNSCC patients had poorer survival than their White counterparts, even after adjusting the analyses for socioeconomic and clinical variables. This suggests a potential difference in disease biology. Chaudhary and colleagues used the cancer genome atlas and cancer digital archive of HNSCC patient and showed that Black patients had a higher frequency of mutations in several genes such as P53 and FATI, and showed less intra-tumoral infiltration of the effector immune cells [31]. Another study comparing mutational differences between Black and White HNSCC patients showed a greater number of copy aberrations and increased miRNA-mediated PRG4 silencing [32]. Future studies should continue to investigate the biological markers and drivers of disease in R/E minorities, which may elucidate opportunities for personalized medicine in these specific populations.

Another unique aspect to our study is that we also evaluated the impact of ethnicity on cancer outcomes. Indeed, it is being increasingly recognized that ethnicity is often ignored in NCDB studies, and in breast cancer, ethnicity was reported in less than 50% of NCDB breast cancer studies [33]. Therefore, our study aimed to disaggregate race and ethnicity so we could better characterize the impact of ethnicity within a given race. Our finding that Hispanic Blacks have better outcomes than non-Hispanic Blacks is a critical finding that could further help with risk stratification as we aim to narrow the disparity gap in HNC.

There are several limitations to this study. First, this is a retrospective descriptive analysis, so selection bias could influence the treatment modality received by each R/E and therefore impact outcomes. Next, data on cancer specific mortality and local control are not available in the NCDB, so it is unclear if the specific cause of death in all patients was due to cancer or other co-morbid illnesses. Next, and importantly, information on smoking status, which can clearly influence disease control outcomes in this population, is not available in the NCDB. Future studies should examine the age of initiation of smoking, potential provider bias in counseling patients on smoking cessation, and rates at which R/E minority patients are referred for smoking cessation services. Another important limitation is the grouping of Asian patients into one group. It is clear that not all Asian patients have similar outcomes and there is clear diversity in geography and outcomes, which has been shown in the breast cancer literature [34].Given the relatively small number of Asian individuals across the large number of head and neck cancer subtypes, we felt that the sample size would be too small for meaningful analysis by country of origin or geographic location (Southeast Asian, South Asian etc.). Finally, data regarding subsequent treatment, including salvage treatment such as immunotherapy, was not available and could impact outcomes in this study.

Therefore, there is an urgent need to create interventions to help minorities and socioeconomically disadvantaged populations overcome obstacles to accessing adequate oncologic treatment. Recently, Manning and colleagues presented data at the American Society of Clinical Oncology (ASCO) 2021 from the Accountability for Cancer Care Through Undoing Racism and Equity (ACCURE) clinical trial and showed that by creating support systems that address race-specific barriers to care, completion rates of treatment improved in Black patients with lung and breast cancer, as did the 5-year survival. The team recently published their data demonstrating that intervention resulted in timely lung cancer surgery for Black patients compared to retrospective cohorts [35]. These are excellent first steps to developing national system-wide interventions to continue to improve outcomes and reduce R/E disparities. NRG Oncology in the last decade has recognized the need to further increase minority enrollment onto clinical trials [36]. However, while increasing minority enrollment in clinical trials is important, implementing systems-based interventions needed to address the root causes of delays in cancer care in a broader setting may even further improve outcomes on a national scale. When survival was adjusted for socioeconomic, clinical, and treatment characteristics in Black patients, the hazard ratio for death improved, but was still not commensurate, suggesting a biologic component in addition to the socioeconomic factors discussed, particularly given the significantly younger age at diagnosis of the R/E minority patients despite the lower rates of HPV-related cancers. Therefore, while systemic interventions to combat health inequity are needed, there is also a role for more funding for basic research to understand the differences in the biology of this disease in R/E minority patients.

## 5. Conclusions

In conclusion, our study demonstrates clear differences in socioeconomic and clinical disease characteristics between different races treated for HNSCC. Black patients had inferior survival outcomes in nearly all head and neck subsites treated. Our data strongly suggest that socioeconomic barriers to care can greatly impact disease outcomes in HNSCC. 

## Figures and Tables

**Figure 1 cancers-15-01781-f001:**
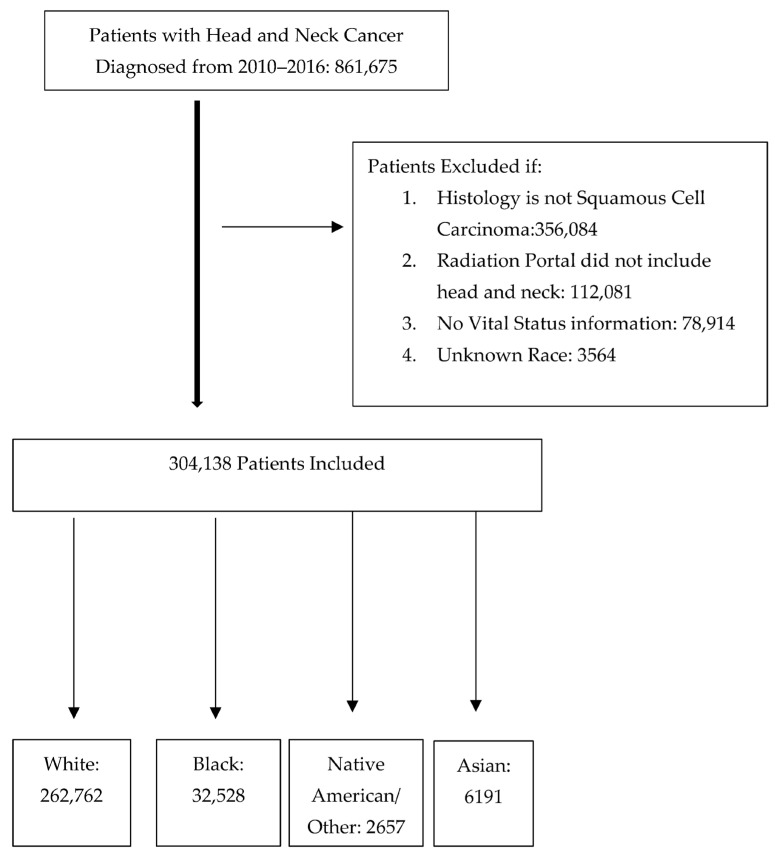
Selection Criteria.

**Figure 2 cancers-15-01781-f002:**
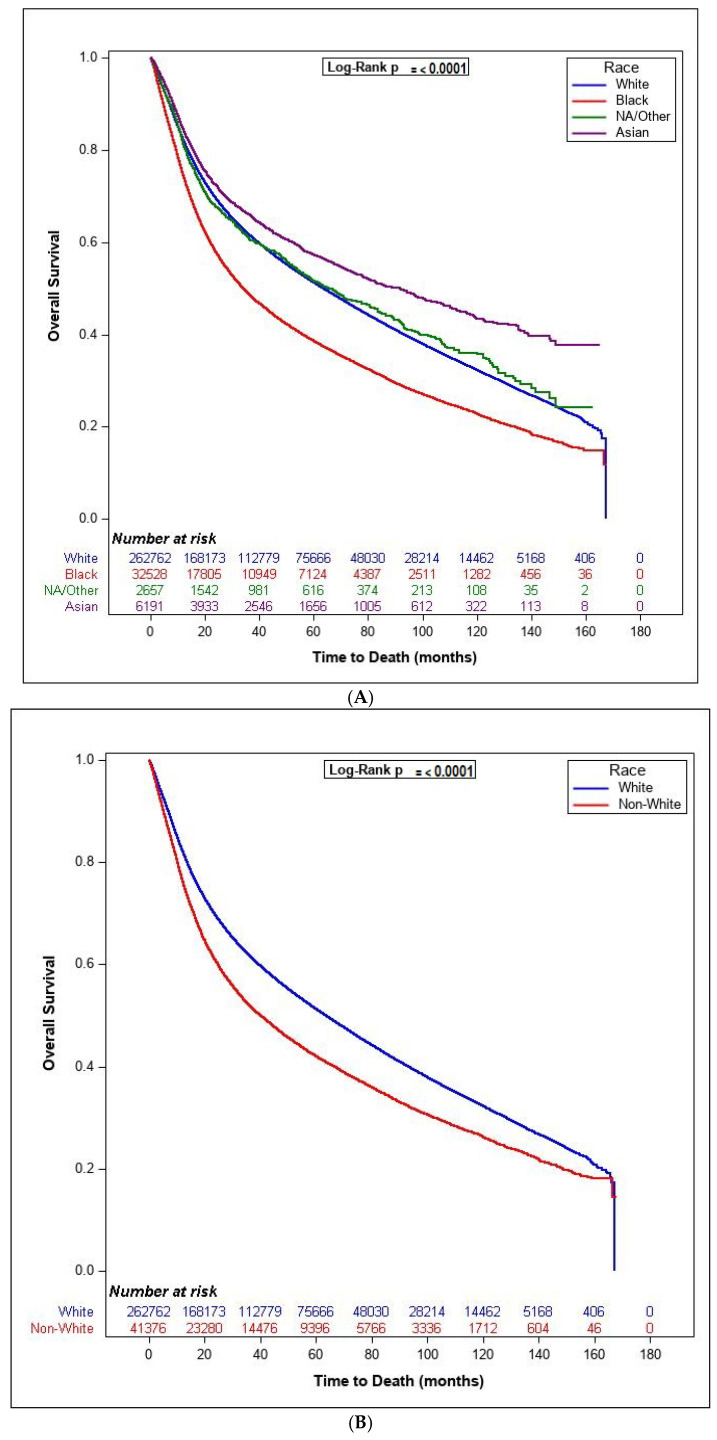
Overall Survival by Racial/Ethnic Minority for (**A**) the entire cohort; (**B**) White vs. non-White; (**C**) Hispanic Black vs. non-Hispanic Black; (**D**) Stage I; (**E**) Stage II; (**F**) Stage III; (**G**) Stage IV.

**Figure 3 cancers-15-01781-f003:**
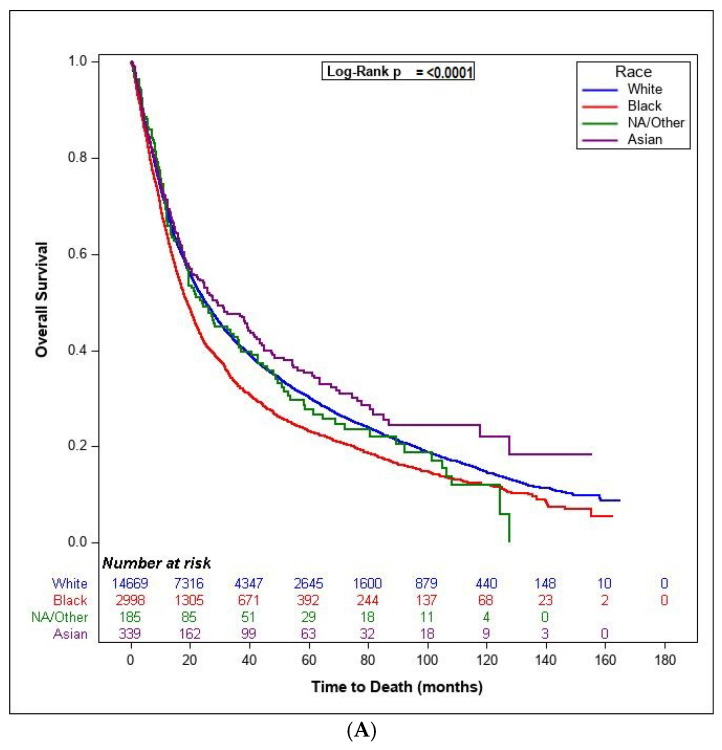
Overall Survival by Racial/Ethnic Minority for the (**A**) Hypopharynx; (**B**) Larynx; (**C**) Nasal Cavity; (**D**) Nasopharynx; (**E**) Oropharynx; (**F**) Pharynx (NOS); (**G**) Oral Cavity.

**Table 1 cancers-15-01781-t001:** Demographic and Clinical Characteristics by Racial/Ethnic Minority.

Variable	Level	White (*n* = 262,762)	Black (*n* = 32,528)	NA/Other (*n* = 2657)	Asian (*n* = 6191)	Total (*n* = 304,138)	*p*-Value *
Age at Diagnosis	Mean (SD) (min, max)	64 (12.2) (18, 90)	61.2 (11.1) (18, 90)	61.4 (12.4) (20, 90)	61 (14.1) (18, 90)	63.6 (12.1) (18, 90)	<0.0001
Site	Oral Cavity	134,941 (51%)	10,714 (33%)	1297 (49%)	3170 (51%)	150,122 (49%)	<0.0001
	Hypopharynx	14,669 (6%)	2998 (9%)	185 (7%)	339 (5%)	18,191 (6%)	
	Larynx	84,446 (32%)	14,203 (44%)	844 (32%)	1112 (18%)	100,605 (33%)	
	Nasal Cavity	7816 (3%)	940 (3%)	82 (3%)	248 (4%)	9086 (3%)	
	Nasopharynx	5622 (2%)	1153 (4%)	115 (4%)	1141 (18%)	8031 (3%)	
	Oropharynx	11,396 (4%)	2000 (6%)	95 (4%)	121 (2%)	13,612 (4%)	
	Pharynx	3872 (1%)	520 (2%)	39 (1%)	60 (1%)	4491 (1%)	
Sex	Female	73,263 (28%)	8255 (25%)	682 (26%)	1899 (31%)	84,099 (28%)	<0.0001
	Male	189,499 (72%)	24,273 (75%)	1975 (74%)	4292 (69%)	220,039 (72%)	
Ethnicity	Hispanic	11,036 (4%)	269 (1%)	553 (23%)	47 (1%)	11,905 (4%)	<0.0001
	Non-Hispanic	238,635 (96%)	30,764 (99%)	1848 (77%)	5958 (99%)	277,205 (96%)	
Insurance	Not Insured	11,968 (5%)	2884 (9%)	207 (8%)	404 (7%)	15,463 (5%)	<0.0001
	Private Insurance/Managed Care	98,657 (38%)	8051 (25%)	903 (34%)	2719 (44%)	110,330 (36%)	
	Medicaid	21,348 (8%)	7111 (22%)	395 (15%)	947 (15%)	29,801 (10%)	
	Medicare	119,955 (46%)	12,766 (39%)	891 (34%)	1943 (31%)	135,555 (45%)	
	Other Government	5279 (2%)	707 (2%)	139 (5%)	44 (1%)	6169 (2%)	
	Insurance Status Unknown	5555 (2%)	1009 (3%)	122 (5%)	134 (2%)	6820 (2%)	
Income	<$30,000	33,005 (13%)	13,263 (42%)	605 (23%)	428 (7%)	47,301 (16%)	<0.0001
	$30,000–$34,999	50,406 (20%)	6856 (22%)	459 (18%)	687 (11%)	58,408 (20%)	
	$35,000–$45,999	75,161 (30%)	6746 (21%)	609 (24%)	1394 (23%)	83,910 (28%)	
	≥$46,000	95,980 (38%)	4876 (15%)	912 (35%)	3489 (58%)	105,257 (36%)	
AJCC Clinical Stage Group (7th Edition)	0	8621 (3%)	735 (2%)	76 (3%)	134 (2%)	9566 (3%)	<0.0001
	1	58,039 (22%)	4323 (13%)	506 (19%)	1226 (20%)	64,094 (21%)	
	2	33,649 (13%)	3708 (11%)	316 (12%)	753 (12%)	38,426 (13%)	
	3	35,741 (14%)	5161 (16%)	396 (15%)	915 (15%)	42,213 (14%)	
	4	76,887 (29%)	12,662 (39%)	827 (31%)	1886 (30%)	92,262 (30%)	
	4C	5187 (2%)	1286 (4%)	77 (3%)	176 (3%)	6726 (2%)	
Surgery	Primary Site only	68,269 (26%)	4906 (15%)	560 (21%)	1367 (22%)	75,102 (25%)	<0.0001
	Primary and Lymph Node	64,837 (25%)	6424 (20%)	740 (28%)	1897 (31%)	73,898 (24%)	
	Lymph Node only	13,080 (5%)	1141 (4%)	125 (5%)	373 (6%)	14,719 (5%)	
	No primary or lymph Node	116,576 (44%)	20,057 (62%)	1232 (46%)	2554 (41%)	140,419 (46%)	
Palliative Care	No palliative care provided	256,845 (98%)	31,377 (96%)	2546 (96%)	6053 (98%)	296,821 (98%)	<0.0001
	Surgery	1046 (0%)	192 (1%)	12 (0%)	27 (0%)	1277 (0%)	
	Radiation Therapy	1391 (1%)	368 (1%)	23 (1%)	24 (0%)	1806 (1%)	
	Chemo/hormone/other systemic drugs	811 (0%)	205 (1%)	12 (0%)	17 (0%)	1045 (0%)	
	Pain management	799 (0%)	117 (0%)	6 (0%)	27 (0%)	949 (0%)	
	Other	947 (0%)	201 (1%)	14 (1%)	41 (1%)	1203 (0%)	
	Unknown	923 (0%)	68 (0%)	44 (2%)	2 (0%)	1037 (0%)	
Charlson–Deyo Score	0	199,731 (76%)	23,922 (74%)	2048 (77%)	4985 (81%)	230,686 (76%)	<0.0001
	1	46,509 (18%)	6131 (19%)	455 (17%)	976 (16%)	54,071 (18%)	
	2	12,084 (5%)	1602 (5%)	116 (4%)	178 (3%)	13,980 (5%)	
	3	4438 (2%)	873 (3%)	38 (1%)	52 (1%)	5401 (2%)	
Metastatic Disease	No	246,892 (98%)	29,638 (95%)	2469 (97%)	5613 (96%)	284,612 (97%)	<0.0001
	Yes	6248 (2%)	1467 (5%)	86 (3%)	222 (4%)	8023 (3%)	
Chemotherapy	No	135,110 (58%)	13,523 (47%)	1323 (56%)	2837 (52%)	152,793 (57%)	<0.0001
	Yes	98,139 (42%)	15,122 (53%)	1032 (44%)	2597 (48%)	116,890 (43%)	
Radiation (RT)	No	99,367 (39%)	9483 (30%)	1017 (39%)	2312 (38%)	112,179 (38%)	<0.0001
	Yes	158,685 (61%)	22,223 (70%)	1563 (61%)	3739 (62%)	186,210 (62%)	
HPV Status	Negative	28,293 (19%)	3717 (21%)	350 (21%)	931 (24%)	33,291 (20%)	<0.0001
	Positive	16,859 (11%)	1128 (6%)	132 (8%)	236 (6%)	18,355 (11%)	
	Unknown	102,171 (69%)	12,586 (72%)	1154 (71%)	2763 (70%)	118,674 (70%)	
Total dose	Median [IQR]	6000 [0, 7000]	6300 [0, 7000]	5412 [0, 7000]	6000 [0, 7000]	6000 [0, 7000]	<0.0001
Great Circle Distance	Median [IQR] (min, max)	12.6 [5, 31.5] (0, 4960.8)	6.5 [3, 16.6] (0, 1938)	12.7 [5, 36.2] (0.1, 2447.2)	7.8 [4, 15.4] (0, 4963.1)	11.6 [5, 29.6] (0, 4963.1)	<0.0001
Surgery to Start of RT interval	<6 weeks	20,359 (42%)	1939 (36%)	162 (33%)	439 (35%)	22,899 (41%)	<0.0001
	≥6 weeks	28,590 (58%)	3515 (64%)	324 (67%)	824 (65%)	33,253 (59%)	
Distance	<4.8	60,996 (23%)	12,761 (39%)	619 (23%)	1972 (32%)	76,348 (25%)	<0.0001
	4.8–29.7	132,913 (51%)	14,741 (45%)	1296 (49%)	3547 (57%)	152,497 (50%)	
	>29.7	68,853 (26%)	5026 (15%)	742 (28%)	672 (11%)	75,293 (25%)	
Total Treatment Time	Median [IQR]	101 [80, 135]	107 [84, 145]	107 [83, 146]	107 [85, 142]	102 [81, 136]	<0.0001

* Significant *p*-values represent that at least one of the pairwise comparisons is statistically significant.

**Table 2 cancers-15-01781-t002:** Univariate and Multivariate Analyses of Predictors of Survival in Patients with HNSCC.

		Univariate		Multivariable	
		HR (95%CL)	*p*-Value	HR (95%CL)	*p*-Value
Age		1.03 (1.02 1.03)	<0.0001	1.02 (1.02 1.03)	<0.0001
Sex	Male	Ref		Ref	
	Female	1.01 (1 1.02)	0.0584	0.97 (0.94 1.01)	0.2732
Race	White	Ref		Ref	
	Asian	0.8 (0.77 0.84)		0.84 (0.74 0.96)	0.0098
	Black	1.41 (1.39 1.43)	<0.0001	1.09 (1.03 1.15)	0.0018
	Native American/Other	0.98 (0.93 1.04)	0.6226	0.95 (0.78 1.15)	0.6157
Ethnicity	Hispanic	Ref		Ref	
	Non-Hispanic	1.05 (1.02 1.08)	0.0003	1.31 (1.19 1.45)	<0.0001
Insurance Status	Private Insurance/Managed Care	Ref		Ref	
	Not Insured	1.8 (1.75 1.84)	<0.0001	1.55 (1.43 1.68)	<0.0001
	Medicaid	2.2 (2.16 2.24)	<0.0001	1.83 (1.73 1.94)	<0.0001
	Other Government	1.69 (1.63 1.76)	<0.0001	1.15 (1.00 1.31)	0.0359
Income	≥$46,000	Ref		Ref	
	$35,000–$45,999	1.18 (1.16 1.19)	<0.0001	1.17 (1.12 1.22)	<0.0001
	$30,000–$34,999	1.28 (1.26 1.3)	<0.0001	1.21 (1.15 1.27)	<0.0001
	<$30,000	1.46 (1.44 1.48)	<0.0001	1.26 (1.21 1.35)	<0.0001
TNM Clinical Stage	1	Ref		Ref	
	2	1.58 (1.55 1.61)	<0.0001	1.53 (1.43 1.64)	<0.0001
	3	1.84 (1.81 1.87)	<0.0001	1.89 (1.77 2.02)	<0.0001
	4	2.29 (2.26 2.33)	<0.0001	2.60 (2.45 2.76)	<0.0001
Surgery	Primary Site Only	Ref		Ref	
	Primary and Lymph Node	1.3 (1.27 1.32)	<0.0001	0.93 (0.88 0.99)	0.0334
	Lymph Node Only	1.12 (1.09 1.15)	<0.0001	0.79 (0.72 0.86)	<0.0001
	No Primary or Lymph Node	1.86 (1.83 1.88)	<0.0001	1.20 (1.13 1.27)	<0.0001
Charlson Score	0	Ref		Ref	
	1	1.38 (1.36 1.4)	<0.0001	1.19 (1.14 1.24)	<0.0001
	2	1.83 (1.79 1.87)	<0.0001	1.40 (1.30 1.50)	<0.0001
	3	2.17 (2.1 2.25)	<0.0001	1.94 (1.75 2.14)	<0.0001
Metastatic Disease	No	Ref		Ref	
	Yes	3.87 (2.78 3.97)	<0.0001	2.40 (2.07 2.79)	<0.0001
HPV Status	Positive	Ref		Ref	
	Negative	1.93 (1.87 2)	<0.00001	2.02 (1.94 2.11)	<0.0001
Distance from Facility	<4.8	Ref		Ref	
	4.8–29.7	0.84 (0.83 0.85)	<0.0001	0.91 (0.87 0.94)	<0.0001
	29.7	0.84 (0.83 0.86)	<0.0001	0.93 (0.88 0.98)	<0.001
Travel Distance		1 (1 1)	<0.0001	1 (1 1)	0.0213

Ref = Reference; HPV:Human Papillomavirus.

## Data Availability

Data is publicly available from the National Cancer Database and American College of Surgeons Website.

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
