# Peer review of "Disparities in Survival Outcomes among Racial/Ethnic Minorities with Head and Neck Squamous Cell Cancer in the United States"

_cancers, 2023, doi:10.3390/cancers15061781_

Round 1

Reviewer 1 Report

In your supplemental table 1 , when you mention Asian vs White / Asian vs Black / Black vs white – what is the reference level of comparison? Please specify the reference levels so the user can interpret the hazard ratios

There are no figure legends. Please provide figure legends to be able to understand the text better

It is unclear as to what variables were considered in the multivariable analysis?

The authors in their discussion reference several studies where similar studies were carried out . The novelty of the current study is hence unclear .

Reviewer 2 Report

The topic addressed by the authors is very relevant and the methodological approach is adequate to answer the research question. 

I have two minor suggestions:

The introduction could be expanded (e.g., contextualize the head and neck cancer burden in our society - Incidence, Mortality, Years of Life Lost, Years Lived with Disability, and Disability-Adjusted Life Years).

The conclusion section should only address direct inferences that can be made based on the results. Some subjects (e.g., future steps following lung cancer initiatives) are better suited to be included in the discussion. 

Reviewer 3 Report

This study showed the association between races and the outcomes of head and neck cancers based on the large database. It is interesting and provides useful information for understanding the prevalence of head and neck cancers. However, some issues should be resolved.

1. Materials and methods: Genetic background has obvious effects on the outcomes of head and neck cancers. People from the West, South, and East of Asia are very different. Therefore, the assignment of all people from Asia to one group is inappropriate. Meanwhile, the reasons of further grouping patients to Black-Hispanic and White-Hispanic groups should be clearly explained.

2. Table 1. The statistic analyses results are confusing. What do these p-values represent for? How did the differences between each pair of means were analyzed? What are the p values for the comparison between each pair of means? For example, authors claimed that “Compared to White patients, Black patients were more likely to be younger at time of diagnosis with a mean of 64 vs 61.2 years ( p<0.0001)”. Does the p-value “<0.0001” in Table1, Line Age at Diagnosis represent the comparison between White patients and Black patients? How about White patients vs Asian patients, White patients vs NA/other?  

3. HPV infection is an important factor for HNSCC patients. How the HPV status is ascertained in patients should be explained in detail in Materials and methods. Different HPV types, high or low risk, may be associated with patients outcomes. In addition, the results of HPV infection in different groups should be reported in Abstract.

4. Authors conclude that Black race is independently associated with worse OS, in part due to socioeconomic, clinical, and treatment related factors. To ensure this conclusion, authors should do further analyses to evaluate OS between high and low income Black patients, among Black patients with or without insurance, and among Black patients with different treatments.  

5. The explanation of some abbreviated words in manuscript should be given, such as RT, AJCC, ASCO, REF. 

Round 2

Reviewer 3 Report

Authors have significantly improved the manuscript.